# Prediction of the Absorption Characteristics of Non-Uniform Acoustic Absorbers with Grazing Flow

**Yang Ou and Yonghui Zhao ***

State Key Laboratory of Mechanics and Control of Mechanical Structures, Nanjing University of Aeronautics and Astronautics, Nanjing 210016, China
* Correspondence: zyhae@nuaa.edu.cn

**Abstract:** In this paper, planar and the cylindrical broadband non-uniform acoustic absorbers were constructed, both of which use broadband absorption units (BAUs) as their building blocks. The impedance boundary Navier–Stokes equation (IBNSE) method was developed to predict the absorption characteristics of the lined duct with non-uniform acoustic absorbers, in which each small piece of perforated plate is acoustically equivalent to a semi-empirical impedance model through the boundary condition. A total of four semi-empirical impedance models were compared under different control parameters. The full Navier–Stokes equation (FNSE) method was used to verify the accuracy of these impedance models. It was found that the IBNSE method with the Goodrich model had the highest prediction accuracy. Finally, the planar and the cylindrical non-uniform acoustic absorbers were constructed through spatial extensions of the BAU. The transmission losses and the absorption coefficients of the rectangular duct–planar acoustic absorber (RDPAA) and annular duct–cylindrical acoustic absorber (ADCAA) systems under grazing flow were predicted, respectively. The results demonstrated that the broadband absorption of the designed non-uniform acoustic absorbers was achieved. The developed IBNSE method with Goodrich model was accurate and computationally efficient, and can be used to predict the absorption characteristics of an acoustically treated duct in the presence of grazing flow.

**Keywords:** broadband absorption unit; planar and cylindrical non-uniform acoustic absorber; semi-empirical impedance model; impedance boundary Navier–Stokes equations method; grazing flow

## 1. Introduction

Noise, as one of the main environmental pollutants, is increasingly affecting our daily lives. Attenuating noise pollution through effective control measures has attracted great interest due to stringent noise control regulations. Conventional micro-perforated plate (MPP) absorbers have been widely used to attenuate narrowband noise. Different materials can be selected to cope with the harsh operating conditions, such as high temperature environments. Using the impedance equations proposed by Maa [1], the acoustic absorption characteristics of MPPs can be predicted, and the desired absorption performance can be achieved by choosing the appropriate structural parameters for the MPP absorbers [2–5].

An absorber with a complex structure by artificial design is called a metamaterial absorber [6]. Based on the metamaterial design, Patel et al. [7] developed a highly efficient, perfect, large angular and ultrawideband solar energy absorber. Ciaburro et al. [8] designed three-layered metamaterial acoustic absorbers based on reused PVC membranes and metal washers. Taking advantage of its sound-absorbing properties, many acoustic absorbers have been developed, such as spatial folding [9,10], Helmholtz resonance [11,12], thin film [13,14] absorbers, etc.

Acoustic absorbers have important applications in the field of acoustic absorption of ducts. The absorber placed in a duct is called an acoustic liner. It is usually composed

of a uniform perforated facesheet over a honeycomb cavity and called a single degree of freedom (SDOF) acoustic liner [15], which typically absorbs acoustic energy in a narrow frequency band dictated by the resonance and antiresonance frequencies of the cavities. A double degree of freedom (2DOF) acoustic liner [16] can offer a higher absorption bandwidth and is capable of covering the necessary source spectrum. To obtain an adjustable acoustic absorption performance, Yan [17] designed a 2DOF honeycomb acoustic liner and demonstrated the feasibility of adjustable acoustic absorption by changing the height of the back cavity. Gautam [18] investigated the acoustic performance of a 2DOF Helmholtz resonator and elaborated on the effect of changing the internal dimensions of the resonating cavity on the underlying acoustic attenuation.

Historically, traditional SDOF and 2DOF acoustic liners have been designed for reduction of noise at a fixed frequency. However, there is a high demand for broadband noise reduction in engineering practice. To this end, the energy of dominant source modes can be made to redistribute into higher order modes which are more easily suppressed by the absorber. This modal conditioning technology is generally achieved by non-uniform acoustic absorption structures. In recent years, the non-uniform acoustic absorption structure, characterized by the spatial variations of the impedance, has received extensive attention. Previous studies demonstrated that the optimized absorption structures with spatially varying impedance offer increased attenuation compared to uniform absorption structures. Several investigators have investigated the possibility of improved absorption structure performance by means of modal redistribution by incorporating a circumferentially or axially segmented absorption structure into the duct. Watson [19] evaluated the acoustic performances of circumferentially segmented duct absorption structures for a range of frequencies and source structures, and concluded that the circumferentially segmented absorption structure gives better broadband performance than the uniform absorption structure and is not particularly sensitive to changes in modal structure of the source. Brown et al. [20] explored a broadband absorption structure by varying the facesheet porosity and the hole diameter for each individual cavity along the axial direction of the duct, while keeping the facesheet thickness and core depth constant. They found that the mixed arrangement of perforated plates with different parameters could broaden the noise absorption bandwidth. Palani et al. [21] designed a new type of non-uniform acoustic absorber which includes a slanted porous septum concept with varying open areas and a MultiFOCA (Multiple FOlded Cavity Absorber) concept. The results demonstrated that the new structure can improve broadband absorption performance. McAlpine et al. [22] proposed a non-uniform axially segmented liner to attenuate fan noise at high supersonic fan speeds. They found that the acoustic energy was scattered into high radial mode orders, which was better absorbed by the liner. Schiller et al. [23] presented a low drag, axially variable depth acoustic absorption structure containing pairs of resonators coupled together by shared inlet volumes just below the facesheet. This type of absorption structure has the potential to achieve the targeted impedance with fewer openings in the facesheet, and therefore less drag than previous designs.

The theoretical analysis methods of the duct with axial [24–26] and circumferential non-uniform [27–30] absorbers have adopted many assumptions and ignored the effects of turbulence and air viscosity, resulting in unreliable predictions in some cases. In view of this, most of the studies on non-uniform acoustic absorbers in ducts were carried out by numerical computation techniques. Schiller et al. [31] compared COMSOL finite element (FE) results of transmission loss of acoustic absorption structure with experimental measurements, and showed that the FE results were highly reliable. Winkler et al. [32] performed an overview of engine liner modeling and a description of the key physical mechanisms. They pointed out that the mid-fidelity tools, such as COMSOL and ACTRAN, are critical enablers for the evaluation and construction of future complex acoustic liners. Scofano et al. [33] used ACTRAN FE code to compute the insertion loss of an acoustic liner for the given duct flow conditions, and identified the need for highly accurate insertion loss modeling. Zhang et al. [34] investigated the response of slit acoustic liners and their

impedance properties under incident waves with different intensities and frequencies, as well as with different grazing flow Mach numbers. They developed a fully predictive impedance eduction technique by solving the compressible Navier–Stokes equations with accurate boundary conditions. The use of mid-fidelity tools allows researchers to explore in more detail the physics of the more sophisticated absorber designs. In particular, a high-fidelity method that is based on large eddy simulation (LES) can also be used to capture the local unsteady flow effects inside the perforation holes under acoustic excitation [35–37]. Since mid-fidelity as well as high-fidelity approaches are based on a direct resolution of the absorption structure perforations and geometry details in the numerical grid, the computational cost is particularly expensive.

A variable-depth absorber contains chambers with different depths tuned for different frequencies. However, this design will result in a higher structure thickness and lower space utilization [21]. In view of this, this paper first develops a kind of broadband non-uniform acoustic absorber with a thin thickness. The basic building block of the developed absorption structures consists of an MPP and detuned cavities with different volumes. Since the desired cavity volumes in the BAU are mainly realized along the MPP surface, the broadband absorption of the BAU structure can be implemented by a smaller thickness compared with the variable-depth design, and there is no surplus space. The IBNSE method that is based on the semi-empirical Goodrich model, was developed to predict the acoustic absorption characteristics of the duct lined by a non-uniform acoustic absorber at grazing flow condition. The accuracy and the computational efficiency of the developed method were verified by FNSE simulations.

## 2. Full Navier–Stokes Equations (FNSE) Method

Figure 1 shows a schematic diagram of a rectangular duct–acoustic absorber (RDAA) system with grazing flow (flow parallel to the facesheet). In Figure 1, all the walls in the RDAA system are assumed to be rigid, while a portion of the bottom wall is replaced with a sample absorber. In this paper, we mainly focus on the prediction method of the attenuation characteristics of the acoustically treated ducts with a grazing flow. The most straightforward way to accomplish this is to solve linearized Navier–Stokes equations using the FE technique, called the full Navier–Stokes equations (FNSE) method.

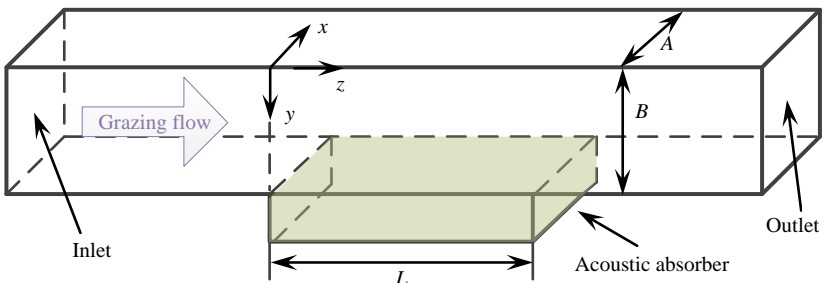

**Figure 1.** Schematic diagram of the rectangular duct–acoustic absorber system under grazing flow.

The linearized Navier–Stokes equations are composed of continuity, momentum and energy equations [38], given by

$$\mathrm{i}\omega\rho + \nabla \cdot (\rho_0 \mathbf{u} + \rho \mathbf{u}_0) = M_0 \tag{1a}$$

$$\rho_0[\mathrm{i}\omega\mathbf{u} + (\mathbf{u}\cdot\nabla)\mathbf{u}_0 + (\mathbf{u}_0\cdot\nabla)\mathbf{u}] + \rho(\mathbf{u}_0\cdot\nabla)\mathbf{u}_0 = \nabla\cdot\boldsymbol{\sigma} + \mathbf{F} - \mathbf{u}_0 M_0 \tag{1b}$$

$$\begin{aligned}\rho_0 C_p(\mathrm{i}\omega T + \mathbf{u}\cdot\nabla T_0 + \mathbf{u}_0\cdot\nabla T) + \rho_0 C_p \mathbf{u}_0\cdot\nabla T_0 \\ -\alpha_p T_0(\mathrm{i}\omega p + \mathbf{u}\cdot\nabla p_0 + \mathbf{u}_0\cdot\nabla p) - \alpha_p T \mathbf{u}_0\cdot\nabla p_0 = \nabla\cdot(k\nabla T) + Q\end{aligned} \tag{1c}$$

where $p$, $\mathbf{u}$, $T$ and $\rho$ are the perturbations of pressure, velocity, temperature and density, respectively. $p_0$, $\mathbf{u}_0$, $T_0$ and $\rho_0$ refer to the steady-state background pressure, velocity, temperature and density, respectively. $\omega$ is the angular frequency of the incident wave, $\mathbf{u}$ is

the vibration velocity of the particle, $k$ is the wave number, $\boldsymbol{\sigma}$ is the stress tensor, and $C_p$ is the constant pressure heat capacity. The mass source term $M_0$, force source term $\mathbf{F}$ and heat source term $Q$ on the right side of equations are zero by default.

The stress tensor and the linearized state equation of an ideal gas [39] are expressed as

$$\boldsymbol{\sigma} = -p\mathbf{I} + \mu[\nabla\mathbf{u} + (\nabla\mathbf{u})^{\mathrm{T}}] + \left(\mu_B - \frac{2}{3}\mu\right)(\nabla \cdot \mathbf{u}_t)\mathbf{I} \tag{2a}$$

$$\rho = \rho_0(\beta_T p - \alpha_p T) \tag{2b}$$

where $\mu$ and $\mu_B$ are the dynamic and bulk viscosity of the material, respectively. $\mathbf{I}$ is the identity matrix. $\alpha_p$ and $\beta_T$ are the isobaric thermal expansion and isothermal compression coefficients, respectively, defined as

$$\alpha_p = -\frac{1}{\rho_0}\left(\frac{\partial\rho_0}{\partial T_0}\right)_p = \frac{1}{c_0}\sqrt{\frac{C_p(\gamma - 1)}{T_0}} \tag{3a}$$

$$\beta_T = \frac{1}{\rho_0}\left(\frac{\partial\rho_0}{\partial p_0}\right)_T = \frac{1}{\rho_0}\frac{\gamma}{c_0^2} = \gamma\beta_S \tag{3b}$$

where $\gamma$ is the specific heat ratio, $c_0$ is the speed of sound in air and $\beta_S = 1/\rho_0 c_0^2$ is the compression coefficients in an adiabatic case. In an adiabatic case, the state equation given by Equation (2b) can be simplified as

$$\rho = \rho_0\beta_S p = \rho_0\frac{\beta_T}{\gamma}p \tag{4}$$

In order to simulate the influences of the background mean flow (grazing flow) on acoustic wave propagation, background mean flow temperature $T_0$, absolute pressure $p_0$ and velocity field $\mathbf{u}_0$ should be defined. In this paper, $T_0$ is taken as 293.15 K, while the absolute pressure and velocity fields are obtained from the turbulent physical field. Assume that there is no tangential stress at the boundary, so the following boundary condition expressions hold true:

$$\mathbf{n} \cdot \mathbf{u} = 0, \ \boldsymbol{\sigma}_n - (\boldsymbol{\sigma}_n \cdot \mathbf{n})\mathbf{n} = 0, \ \boldsymbol{\sigma}_n = \boldsymbol{\sigma}\mathbf{n} \tag{5}$$

where $\mathbf{n}$ is the normal unit vector.

The background mean flow is modeled with an SST RANS model from the CFD Module in COMSOL Multiphysics. This allows us to resolve the boundary layer details. The fluid is treated as a compressible flow. A velocity boundary condition is applied at the inlet, and the velocity of the incoming flow can be written as

$$U_{\mathrm{in}} = M \cdot c_0 \tag{6}$$

where $M$ is the Mach number of the incoming flow.

The pressure boundary condition and reflux inhibition are applied at the outlet and the initial pressure value is set as 0. The wall boundary condition given in Equation (5) is adopted at the other boundaries. First, according to the given Mach number of the incoming flow, the turbulence equations are solved to obtain the background average velocity, average pressure and dynamic viscosity of air. Then, the values of these three background parameters are assigned to the linearized Navier–Stokes physical field to calculate the acoustic pressure field. During simulations, the acoustic wave incidence is parallel to the grazing flow direction, and the area near the inlet is set as the background pressure field. The areas at both ends of the duct are set as perfect matching layers to simulate the far-field situation.

The average acoustic pressure at the inlet and outlet planes can be obtained from the calculation results. Thus, the transmission loss and absorption coefficient can be calculated by Equations (7) and (8), respectively [15].

$$\text{TL} = 20\lg\left|\frac{p_b}{p_t}\right| \tag{7}$$

$$\alpha = 1 - \left|\frac{p_t}{p_b}\right|^2 - \left|\frac{p_s}{p_b}\right|^2 \tag{8}$$

where $p_b$ is the background average acoustic pressure at the inlet plane, $p_s$ is the reflected average acoustic pressure at the inlet plane and $p_t$ is the transmitted average acoustic pressure at the outlet plane.

## 3. Impedance Boundary Navier–Stokes Equations (IBNSE) Method

Since the FNSE method requires detailed FE modeling, including detailed modeling and meshing of micro holes, the computational cost is particularly expensive. In acoustics, the absorber under grazing flow can be replaced by using the concept of impedance. In view of this, the impedance boundary Navier–Stokes equations (IBNSE) method was developed, in which the semi-empirical impedance equations are applied as interior boundary conditions to translate physical parameters of the facesheet into normalized resistance and reactance, and then a set of linearized Navier–Stokes equations are solved. Subsequent simulations show that compared with the FNSE method, the developed IBNSE method is more efficient in computation, and the memory requirement is greatly reduced.

### 3.1. Semi-Empirical Impedance Models of the Perforated Plate under Grazing Flow

The transfer impedance of the perforated plate under the grazing flow can be computed by the so-called semi-empirical impedance model. In this subsection, a total of four impedance models will be introduced. All of them were validated in sound pressure levels of 120~140 dB.

#### 3.1.1. Eversman Model

This semi-empirical impedance model of the perforated plate was proposed by Eversman [40], which is suitable for single degree of freedom and double degree of freedom acoustic liners, and the highest Mach number used in the original simulation was 0.4. The model is expressed as

$$Z_E = R_0 + S_r V + \mathrm{i}\frac{\omega}{c_0\sigma}\left[k_6 t + k_5 0.85\frac{(1 - 0.7\sqrt{\sigma})d}{1 + 305M^3}\right] + R_g\left(\frac{M}{\sigma}, \frac{t}{d}, f\right) \tag{9}$$

where $V$ is the particle velocity magnitude near the holes of the perforated plate, with the correction factors $k_5 = k_6 = 0.9$ and $C_D = 0.88$. $d$ is the hole diameter, $t$ is the thickness of the plate, $\sigma$ is the perforation rate, $f$ is the frequency of incident wave, $k$ is the wave number and $R_0$ is the frequency-independent part of linear resistance.

In Equation (9), the nonlinear acoustic resistance slope $S_r$ and the acoustic impedance $R_0$ are given by

$$S_r = 1.2823\frac{1 - \sigma^2}{2c(C_D\sigma)^2} - 0.0004 \tag{10}$$

$$R_0 = 58.72\frac{\mu}{\rho_0 c_0}\frac{1}{\sigma}\frac{t}{d^2} + 0.0065 \tag{11}$$

$R_g$ in Equation (9) is given by

$$R_g\left(\frac{M}{\sigma}, \frac{t}{d}, f\right) = k_{2i}\left(\frac{145}{\text{OASPL}}\right)^3 f_2(M)\frac{M(5 - t/d)}{4\sigma} - k_{3i}f_3(M)\frac{df}{c\sigma} \tag{12}$$

$$f_2(M) = 0.4 - 0.1M, \ f_3(M) = -0.095 + 2.87M - 3.5M^2 \tag{13}$$

where OASPL is the total acoustic pressure level and the coefficients $k_{2i} = 1.15$ and $k_{3i} = 1.10$.

### 3.1.2. Guess Model

This model was proposed by Guess [41]. The Guess model includes not only the standard impedance terms due to viscosity, radiation and backing effects, but also the terms due to high sound amplitude and steady tangential airflow. The highest Mach number used in the original simulation was 0.2. In this model, the real and imaginary parts of the perforated plate impedance are given by

$$R_G = \frac{\rho_0 c_0}{\sigma} \left[ \frac{\sqrt{8\mu\omega}}{c_0} \left(1 + \frac{t}{d}\right) + \frac{8\mu}{dc_0} + \frac{\pi^2}{2}\left(\frac{d}{\lambda}\right)^2 \right] + \frac{1 - \sigma^2}{\sigma}(\sigma M_0 + 0.31M) \tag{14a}$$

$$X_G = \frac{\rho_0 \omega}{\sigma} \left[ 1 + \frac{8d}{3\pi} \frac{(1 - 0.7\sqrt{\sigma})}{(1 + 305M^3)} \frac{(1 + (5.10)^3 M_0^2)}{(1 + 10^4 M_0^2)} \right] \tag{14b}$$

where $\lambda$ is the acoustic wavelength and $M_0$ is the Mach number near the holes.

### 3.1.3. Lee Model

The Lee model was proposed by Lee and Ih [42]. It can predict the acoustic impedance of circular holes under average flow. The highest Mach number used in the original simulation was 0.2. In the Lee model, the real and imaginary parts of impedance of the perforated plate are given by

$$R_L = a_0(1 + a_1|f - f_0|)(1 + a_2M)(1 + a_3d)(1 + a_4t)/\sigma \tag{15a}$$

$$X_L = b_0(1 + b_1d)(1 + b_2t)(1 + b_3M)(1 + b_4f)/\sigma \tag{15b}$$

where

$$f_0 = 412(1 + 104M)/(1 + 274d), \ a_0 = 3.94 \times 10^{-4}, \ a_1 = 7.84 \times 10^{-3}, \ a_2 = 14.9, \ a_3 = 296 \tag{16a}$$

$$a_4 = -127, \ b_0 = -6 \times 10^{-3}, \ b_1 = 194, \ b_2 = 432, \ b_3 = -1.72, \ b_4 = -6.62 \times 10^{-3} \tag{16b}$$

### 3.1.4. Goodrich Model

A representative semi-empirical model for predicting the absorber impedance is the Goodrich impedance model proposed by the Goodrich Aeronautical Structure Group [43]. The highest Mach number used in the original simulation was 0.2. In the Goodrich model, the frequency normalized linear acoustic impedance of the perforated plate is calculated as follows:

$$Z_G = Z_{of} + S_r V + R_{cm} + i(S_m V) \tag{17}$$

where the impedance of the perforated plate $Z_{of}$ can be expressed as

$$Z_{of} = i\omega \frac{t + \varepsilon d}{c_0 \sigma} F(k_s r) \tag{18}$$

where the end correction $\varepsilon d$ is

$$\varepsilon d = (0.2d + 0.85id)(1 - 0.7\sqrt{\sigma}) \tag{19}$$

The cross-section averaged hole velocity profile $F(k_s r)$ is

$$F(k_s r) = 1 - \{2J_1(k_s r)/[k_s r J_0(k_s r)]\}, \ k_s^2 = -i\omega\rho_0/\mu \tag{20}$$

The nonlinear acoustic resistance slope $S_r$, normalized grazing flow acoustic resistance $R_{cm}$ and nonlinear mass acoustic reactance slope $S_m$ are expressed as

$$S_r = \frac{1.336541}{\rho c}\left(\frac{\rho_0}{2C_d^2}\frac{1-\sigma^2}{\sigma^2}\right), \; R_{cm} = \frac{M}{\sigma\left(2 + 1.256\frac{\delta^*}{d}\right)}, \; S_m = -0.0000207\frac{k}{\sigma^2} \tag{21}$$

For micro-perforates, the flow coefficient $C_d$ is

$$C_d = 0.584854\sqrt{\sigma^{0.1}/e^{-1.151d/t}} \tag{22}$$

### 3.2. Comparisons of Different Impedance Models

The impedance characteristics of the perforated plate were calculated using the above four semi-empirical models. The real part $\text{Re}(z_M)$ and imaginary part $\text{Im}(z_M)$ of the relative specific surface impedance of perforated plate $z_M$ (unit: 1) were compared as a function of hole diameter, perforation rate, Mach number and frequency. The relative surface impedance $z_M$ is expressed as

$$z_M = \frac{S_0}{\rho_0 c_0}Z_M \tag{23}$$

where $S_0$ is the area of the MPP and $Z_M$ is the impedance of the perforated plate.

The basic parameters used for the comparison are $d = 0.8$ mm, $\sigma = 8.04\%$, $M = 0.1$ and $f = 3000$ Hz. The acoustic pressure level was set as 120 dB. According to the report from the MPP manufacturers [44], the hole diameter and the panel thickness are usually equal due to manufacturing limitations. Therefore, in this study, the hole diameter $d$ and the thickness of the MPP $t$ were set to $t = d$. In each round of simulation, only one of the above four parameters was allowed to change, and the other parameters remain unchanged. The comparison results are shown in Figure 2.

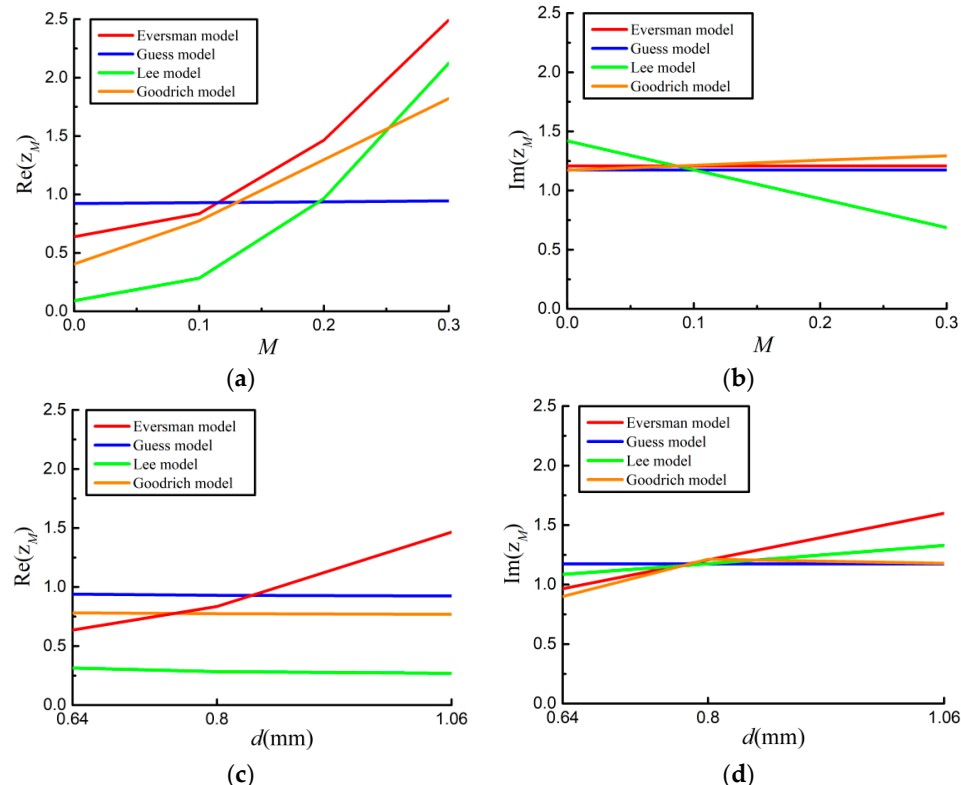

**Figure 2.** *Cont.*

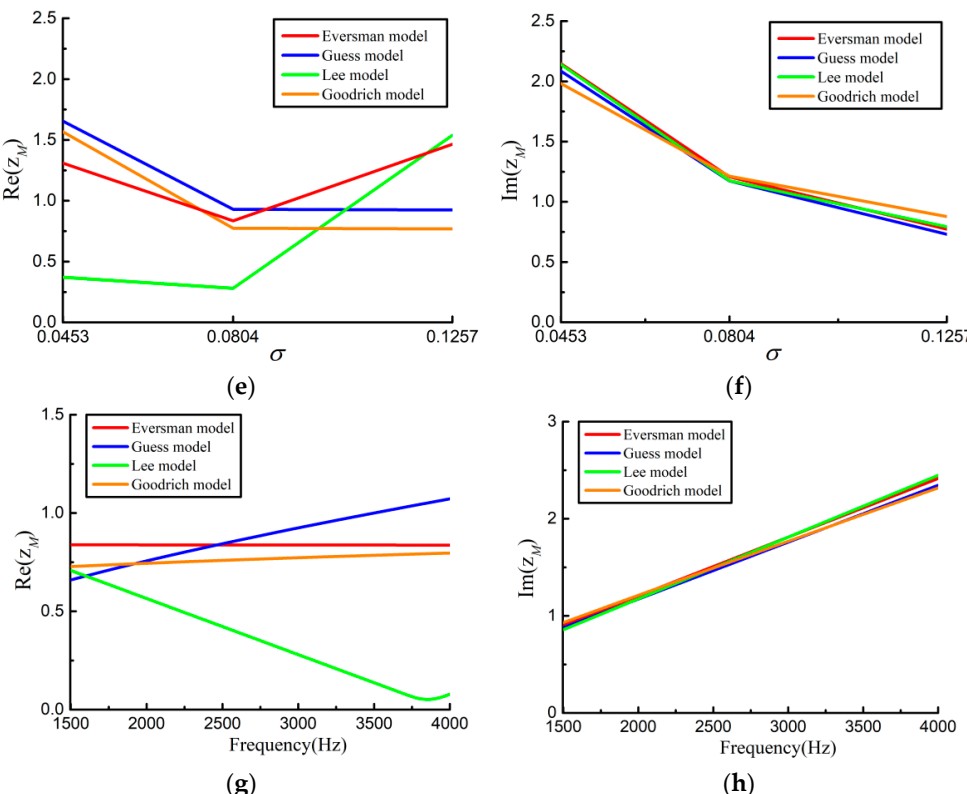

**Figure 2.** Comparisons of different impedance models. (**a**) Real part of impedance vs. Mach number; (**b**) imaginary part of impedance vs. Mach number; (**c**) real part of impedance vs. hole diameter; (**d**) imaginary part of impedance vs. hole diameter; (**e**) real part of impedance vs. perforation rate; (**f**) imaginary part of impedance vs. perforation rate; (**g**) real part of impedance vs. Frequency; (**h**) imaginary part of impedance vs. frequency.

It can be seen from Figure 2 that the impedance characteristics predicted by the four semi-empirical models were very different, especially the real part of the impedance. In the influence of frequency on impedance, the four models gave very similar results for the imaginary part of impedance. The sound absorption performance of the absorber depends on the sum of the MPP impedance and the cavity impedance. When the real part of the sum of impedances is 1 and the imaginary part is 0, the theoretical sound absorption coefficient reaches 1 and, in this case, the perfect sound absorption occurs. Since there is no exact theoretical formula for the impedance of complex cavities in grazing flow, the sound absorption characteristics of the whole structure can be calculated by combining simulation of cavities and a semi-empirical theoretical model of MPP. The next section will show that the proper selection of the semi-empirical impedance models in the IBNSE method can produce an accurate prediction of the transmission loss characteristics of the acoustically treated duct system under grazing flow.

### 3.3. IBNSE Method Using Semi-Empirical Impedance Model

Different from the FNSE method, the developed IBNSE method uses the transfer impedance to replace the perforated plates with an impedance plane. The computational flow chart of the IBNSE method is shown in Figure 3.

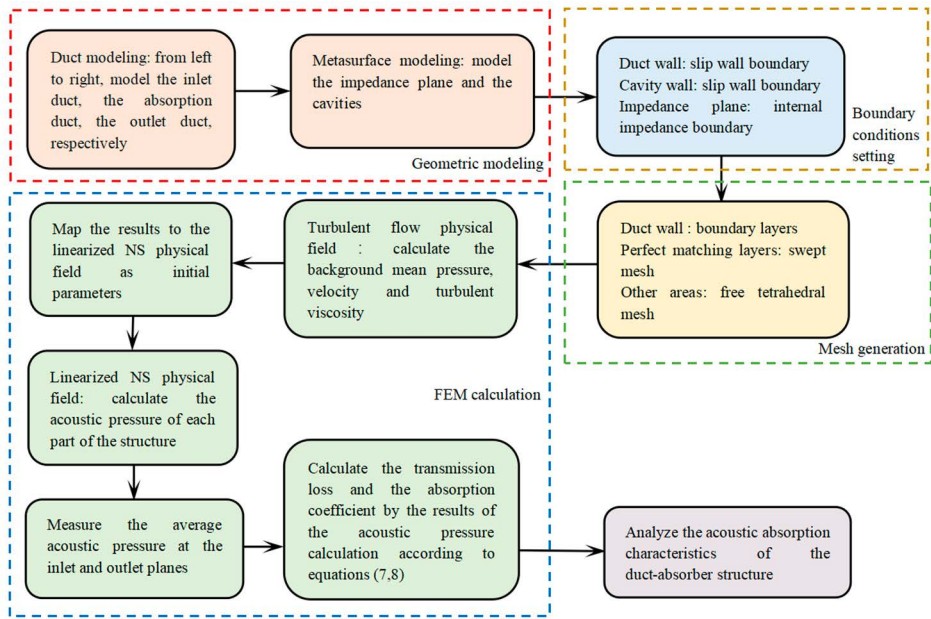

**Figure 3.** Computational flow chart of the impedance boundary Navier–Stokes equations method.

In the IBNSE method, the impedance boundary conditions satisfy the following equations

$$\boldsymbol{\sigma}_{\text{up}} - \boldsymbol{\sigma}_{\text{down}} = -(p_{\text{up}} - p_{\text{down}})\mathbf{n}, \ p_{\text{up}} - p_{\text{down}} = -Z_M(\mathbf{n} \cdot \mathbf{u}) \tag{24}$$

where $\boldsymbol{\sigma}_{\text{up}}$ and $\boldsymbol{\sigma}_{\text{down}}$ are the stress tensor of both sides of the perforated plate and $p_{\text{up}}$ and $p_{\text{down}}$ are the pressures of both sides of the perforated plate, respectively. Since the detailed modeling and meshing of micro-perforated plates are omitted, the computational efficiency of the IBNSE will be greatly increased.

In order to determine which semi-empirical model is the most accurate, a basic resonance element (BRE) was first constructed, which includes an MPP and a backing cavity, as shown in Figure 4. The micro holes reflect acoustic resistance, while the cavity contributes to acoustic reactance, so the combination of MPP and the cavity works as an acoustic absorber. In Figure 4, the width of the MPP is $a = 10$ mm and the length of the MPP and the cavity is $h$. Note that $h$ is a variable parameter used to adjust the volume of the cavity. Since the cavity length is extended along the MPP surface, different cavity volumes can be designed to produce different resonance frequencies without increasing the thickness of structure. The thickness of the MPP is $t$ and all the wall thicknesses of the cavity are $b = 1$ mm. The width and the height (thickness) of the cavity are the same as the width of MPP. The number of micro holes is $n$, the hole diameter is $d$ and the perforation rate of MPP is $\sigma$. The air density is $\rho_0 = 1.2$ kg/m$^3$, sound velocity is $c_0 = 343$ m/s and dynamic viscosity coefficient is $\mu = 1.8 \times 10^5$ Pa $\cdot$ s.

Next, the FNSE and IBNSE methods were used to verify the accuracy of these semi-empirical impedance models. To this end, a basic resonance element (BRE) was placed on the lower wall of the rectangular duct, forming a rectangular duct–BRE (RDBRE) system. Referring to Figure 1, the length and width of the duct section are $A = B = 12$ mm, the total length is $L_t = 80$ mm and the length of the cavity $h$ in Figure 4 was set as 15 mm.

In this validation process, the grazing flow Mach number was taken as 0.1 and the incident acoustic pressure level was 120 dB. The acoustic attenuation characteristics of the BRE absorber under grazing flow were computed by the FNSE and IBNSE methods, respectively. In the FNSE simulations, all the detail structures including all micro-holes and backing cavities in the BRE should be modeled and meshed, as shown in Figure 5. Boundary layers were used on the duct wall and hole wall. The number of layers was 6 and the stretch factor of the boundary layers was 1.2. The free tetrahedral grid was used for

the rest of the structure. For the IBNSE method, by contrast, it is not necessary to establish geometric and FE models of the MPP since the acoustic characteristic of the MPP under grazing flow is replaced by the semi-empirical transfer impedance model, as shown in Figure 6. To generate the perfect matching layer, a swept mesh was adopted at both ends of the duct.

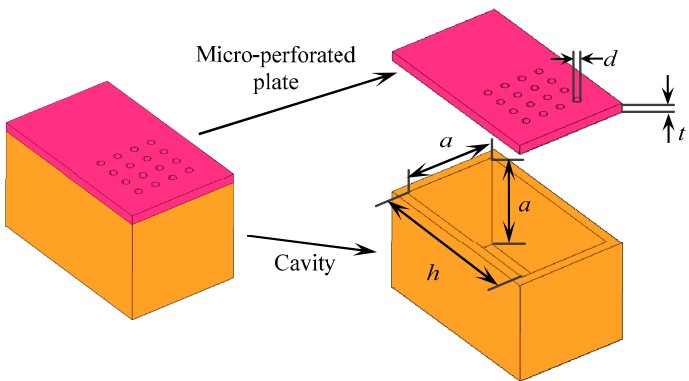

**Figure 4.** Schematic view of the basic resonance element.

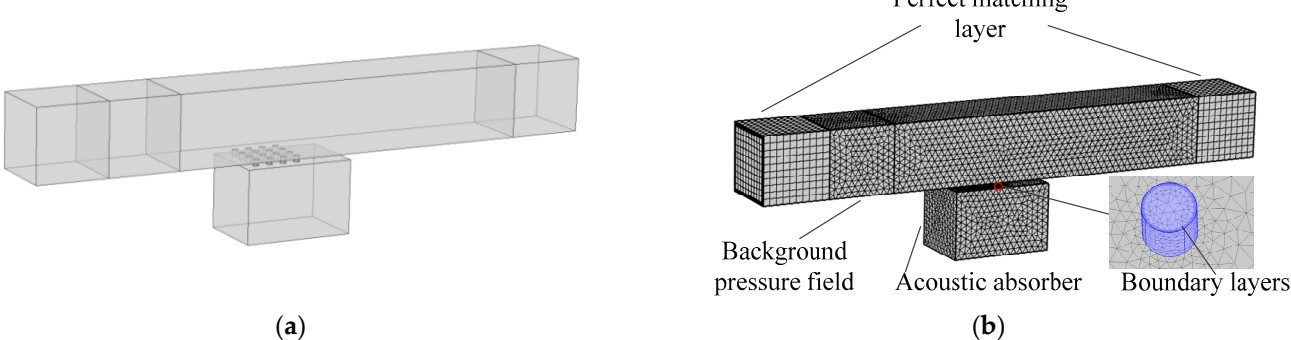

**Figure 5.** RDBRE system with grazing flow (suitable for the FNSE method). (**a**) geometric model. and (**b**) FE model.

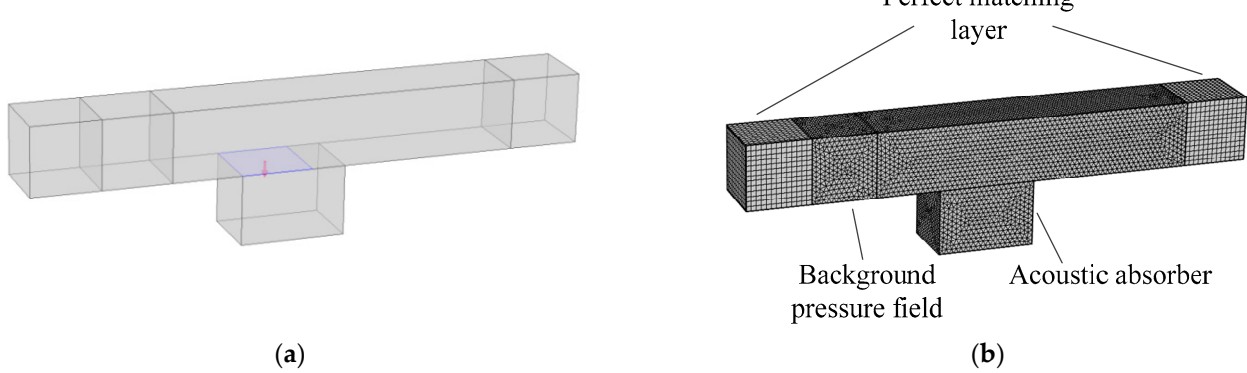

**Figure 6.** RDBRE system with grazing flow (suitable for the IBNSE method). (**a**) Geometric model. and (**b**) FE model.

The parameters given in Section 3.2 were used to calculate the transmission loss using the IBNSE method with different impedance models. The prediction results were compared with those obtained by the FNSE method, as shown in Figure 7. The average errors of the predicted transmission loss obtained by the IBNSE method are listed in Table 1, which were obtained by calculating the average value of the difference of transmission losses obtained by the IBNSE and the FNSE methods. As demonstrated in Figure 7 and Table 1, the IBNSE

method using the Goodrich model had the highest accuracy for the different parameter variations. Therefore, the Goodrich model was used for calculations in the following IBNSE simulations.

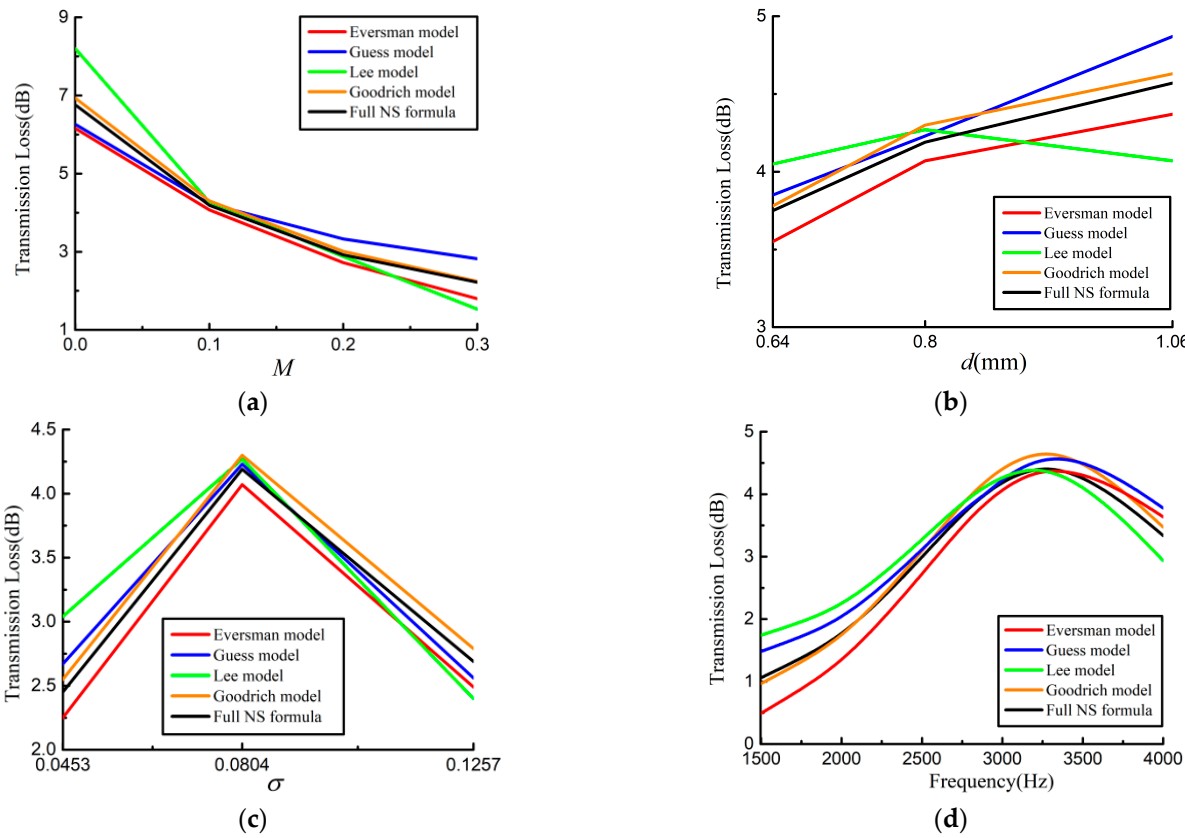

**Figure 7.** Comparisons of transmission loss using different impedance models. (**a**) Different Mach number; (**b**) different hole diameters; (**c**) different perforation rates; (**d**) different frequencies.

**Table 1.** Average relative errors of the predicted transmission loss using the IBNSE method under different parameter variations (Mach number, hole diameter, perforation rate and frequency).

| Impedance Models | Mach Number $M$ | Hole Diameter $d$ | Perforation Rate $\sigma$ | Frequency $f$ |
|---|---|---|---|---|
| Eversman model | 8.38% | 4.22% | 5.65% | 5.86% |
| Guess model | 9.68% | 3.59% | 3.77% | 8.86% |
| Lee model | 14.07% | 7.15% | 10.32% | 11.66% |
| Goodrich model | 2.45% | 1.63% | 3.22% | 3.28% |

Next, the influences of the MPP parameters on the acoustic attenuation performance were studied using the IBNSE method with Goodrich model and the FNSE method. In simulations, let the hole diameter $d$ and the perforation rate $\sigma$ in the four basic parameters given in Section 3.2 be the variable parameters, and the other three parameters are fixed. In Figure 8a, three MPP cases with hole diameters of 0.64, 0.8 and 1.06 mm were considered. Correspondingly, the number of holes $n$ was changed to 25, 16 and 9, so that the perforation rate $\sigma$ remains unchanged at 8.04%. In Figure 8b, three MPP cases with perforation rates of 4.52%, 8.04% and 12.57% were used, and the number of holes $n$ was changed accordingly to 9, 16 and 25, so that the hole diameter $d$ remains unchanged at 0.8 mm.

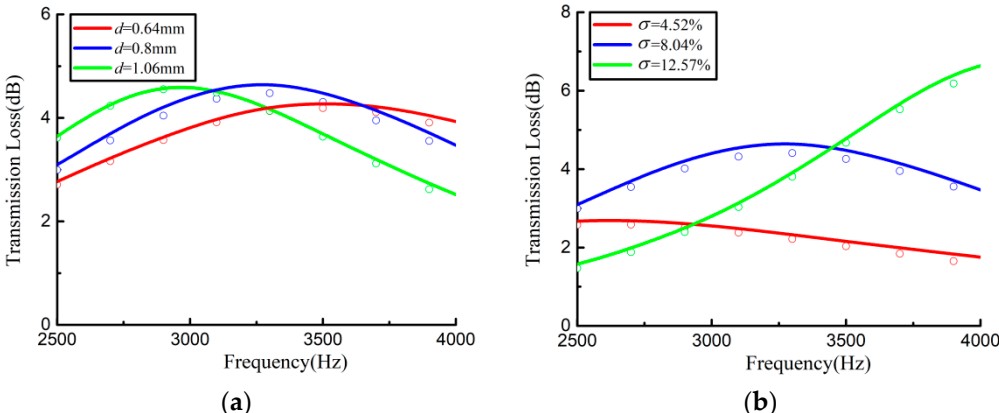

**Figure 8.** Transmission loss predicted using different perforated plate parameters. The solid lines represent the results computed by the IBNSE method with Goodrich model, while the circles represent the results computed by the FNSE method. (**a**) Different hole diameters; (**b**) different perforation rates.

From Figure 8a, it can be seen that a larger hole diameter produced a higher peak of transmission loss, and the peak frequency decreased with the increase in hole diameter. Figure 8b shows that the amplitude and frequency of the transmission loss peak increased with the increase in perforation rate. Within the given frequency band, the IBNSE method yielded accurate results using different perforated plate parameters.

In order to further verify the accuracy of the Goodrich model, simulations are carried out under the perforation parameters $d = 0.8$ mm and $\sigma = 8.04\%$. The transmission losses were predicted by the IBNSE method with Goodrich model under different Mach numbers and compared with the results obtained by the FNSE method, as shown in Figure 9. Again, the transmission loss at different Mach numbers obtained by the IBNSE method with Goodrich model were in good agreement with those computed by the FNSE method. In addition, it can be found that the peak value of transmission loss decreased with the increase of the Mach number. This is because the high grazing flow velocity increases the acoustic resistance of the overall structure through $R_{cm}$ in Equation (21), the total relative surface resistance has a larger difference with respect to 1 and the transmission loss is roughly inversely proportional to the absolute value of the total impedance of the absorber [45].

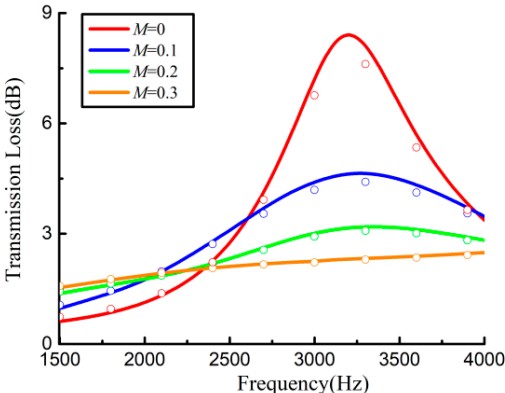

**Figure 9.** Transmission loss of the RDBRE system with different grazing flow Mach numbers. The solid lines represent the results computed by the IBNSE method with Goodrich model, while the circles represent the results computed by the FNSE method.

### 3.4. Comparisons with Published Experimental Results

In order to further verify the accuracy of the IBNSE method in this paper, the absorber given in Reference [15] was used, and the transmission loss and acoustic absorption coefficient of the absorber were calculated by FNSE method and IBNSE method, respectively.

The acoustic absorber is composed of a number of honeycomb cavities connected with a micro-perforated plate. As shown in Figure 10, the height of the cavities is $h = 25$ mm, the side length of the cavity section is $a = 2.8$ mm and the hole diameter and the thickness of MPP are $d = t = 0.5$ mm. The cross section of the duct is rectangular, and corresponding the length and width are $A = B = 100$ mm. The total length of the duct is 950 mm and the length of the acoustic liner part is $L = 500$ mm. The Mach number of the grazing flow is 0.035. The geometric and FE models of the structure suitable for FNSE and IBNSE methods are shown in Figures 11 and 12, respectively.

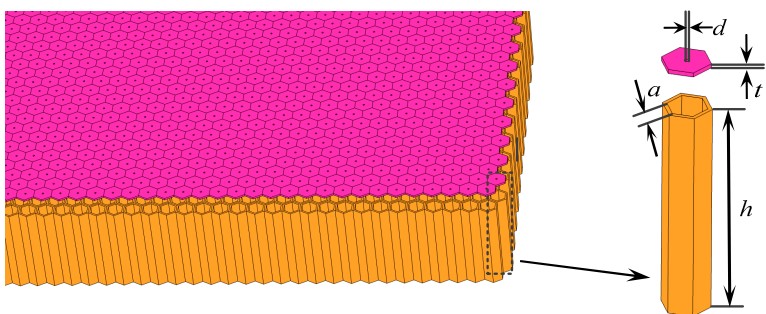

**Figure 10.** Schematic view of the structure in Reference [15].

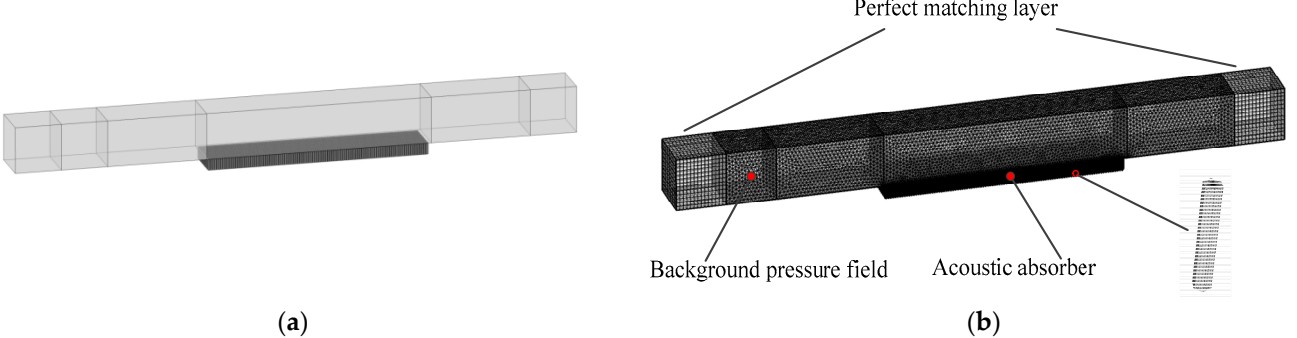

(**a**)                                                    (**b**)

**Figure 11.** Structure in Reference [15] (suitable for the FNSE method). (**a**) Geometric model and (**b**) FE model.

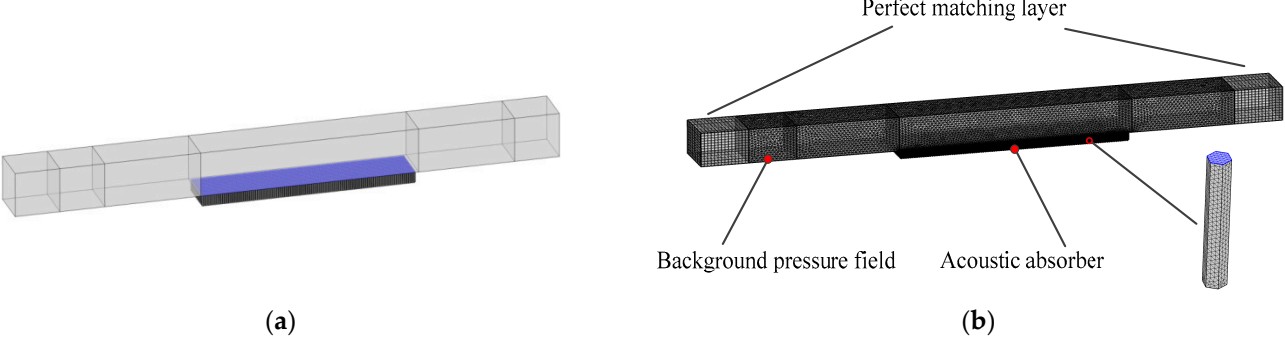

(**a**)                                                    (**b**)

**Figure 12.** Structure in Reference [15] (suitable for the IBNSE method). (**a**) Geometric model and (**b**) FE model.

Figure 13 reveals that under different Mach numbers, the transmission loss and absorption coefficient obtained by the IBNSE method agreed very well with those obtained by the FNSE method. In addition, we can see that these calculated results are in good agreement with the experimental measurements, which once again proves the effectiveness of the developed IBNSE method.

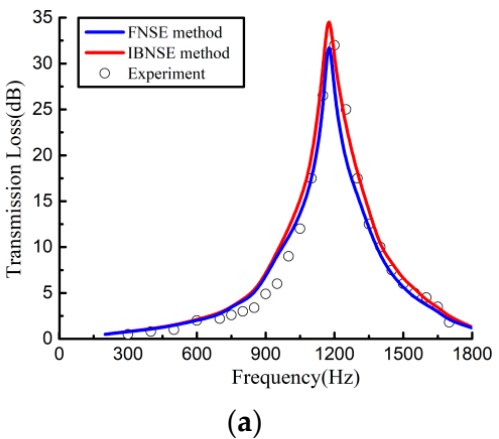

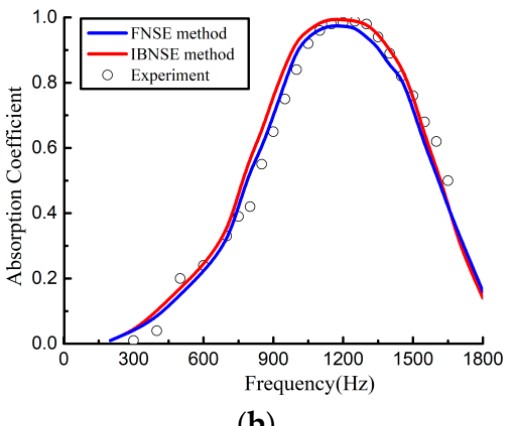

(**a**)

(**b**)

**Figure 13.** Comparisons of transmission loss and absorption coefficient of the structure in Reference [15]. (**a**) Transmission loss and (**b**) acoustic absorption coefficient.

Tables 2 and 3 show the errors of the predicted transmission loss and absorption coefficient obtained by the IBNSE method and the FNSE method compared to the experimental results given by Reference [15]. As demonstrated in Tables 2 and 3, the IBNSE method proposed in this paper had satisfactory computational accuracy.

**Table 2.** Transmission loss errors corresponding to the FNSE and the IBNSE methods.

| Methods | Average Error | Peak Frequency Error | Peak Value Error |
|---------|---------------|----------------------|------------------|
| IBNSE | 5.87% | 1.75% | 4.69% |
| FNSE | 4.24% | 1.67% | 2.50% |

**Table 3.** Absorption coefficient errors corresponding to the FNSE and the IBNSE methods.

| Methods | Average Error | Peak Frequency Error | Peak Value Error |
|---------|---------------|----------------------|------------------|
| IBNSE | 4.06% | 2.48% | 0.71% |
| FNSE | 3.89% | 2.40% | 1.32% |

## 4. Absorption Characteristics of the Duct–Acoustic Absorber System

In this section, a broadband absorption unit (BAU) was constructed. Then, planar and the cylindrical non-uniform acoustic absorbers were formed by extension of the BAU in space. The absorption characteristics of these two absorbers were computed using the developed IBNSE method, in which the Goodrich model was used to obtain the transfer impedance of the lined segments.

### 4.1. Rectangular Duct Acoustic Absorber

As shown in Figure 4, the cavity length $h$ dominated the imaginary part of the overall impedance and hence determined the resonance frequency. Thus, the basic resonant elements with different cavity lengths can be combined together to generate the desired resonance frequency distribution and expand the absorption bandwidth. To this end, a broadband absorption unit (BAU) absorber consisting of six BREs with the same cavity thickness $a = 10$ mm and different cavity lengths was constructed, as shown in Figure 14. The six cavity lengths are $h_1 = 28$ mm, $h_2 = 21$ mm, $h_3 = 20$ mm, $h_4 = 19$ mm, $h_5 = 18$ mm and $h_6 = 11$ mm. They were determined by the maximum average transmission loss of the BAU absorber with a large number of different cavity lengths. The hole diameter $d = 0.8$ mm and perforation rate $\sigma = 8.04\%$ were used in the following simulations.

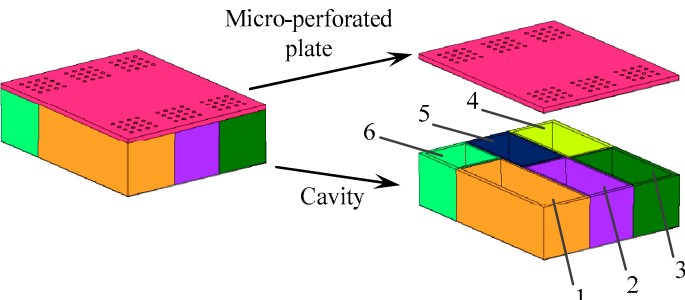

**Figure 14.** Schematic view of the planar BAU.

For a practical design of a duct acoustic absorber system, the acoustically treated area in the duct should be large enough, which can be achieved by increasing the number of BAUs. To this end, a planar non-uniform acoustic absorber consisting of 20 BAUs was constructed, as shown in Figure 15.

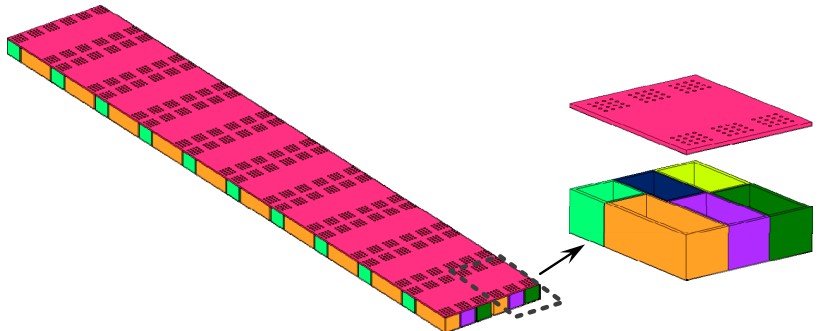

**Figure 15.** Planar non-uniform acoustic absorber formed by planar BAUs.

Referring to Figure 1, the designed planar non-uniform acoustic absorber was placed in the rectangular duct as an acoustically treated liner. In simulations, the length and width of the duct section were $A_1 = B_1 = 68$ mm, the total length of the duct was $L_{\text{duct}} = 810$ mm and the length of the planar acoustic absorber was $L = 410$ mm. The incident acoustic pressure level was taken as 120 dB. Again, for comparison purposes, both the FNSE and the IBNSE methods were used in simulations. Figures 16 and 17 show the geometric and FE models of the rectangular duct–planar acoustic absorber (RDPAA) system suitable for the FNSE method and IBNSE method, respectively.

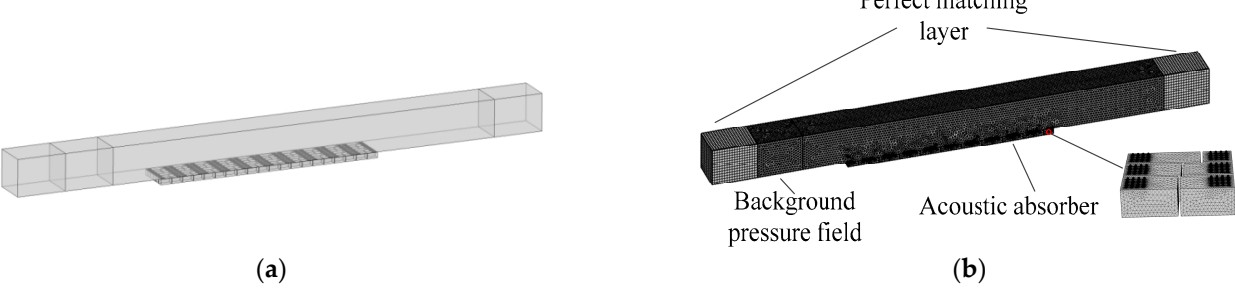

(**a**)                                                                                              (**b**)

**Figure 16.** Geometric and detailed FE models of the RDPAA system with grazing flow (suitable for the FNSE method). (**a**) Geometric model and (**b**) FE model.

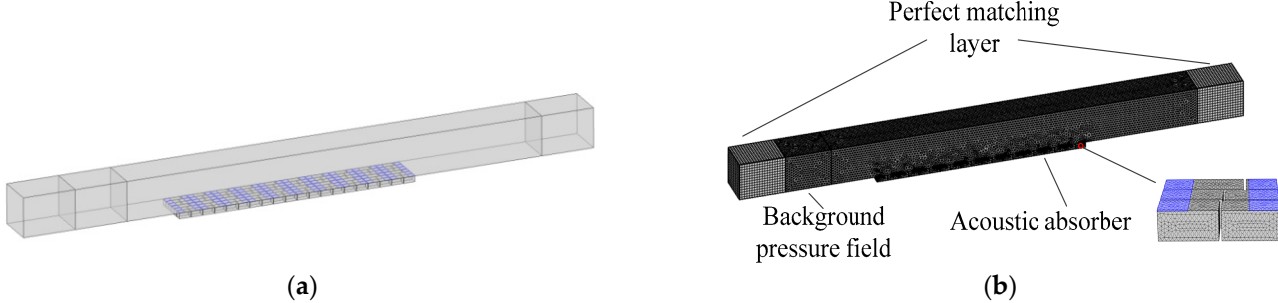

**Figure 17.** Geometric and detailed FE models of the RDPAA system with grazing flow (suitable for the IBNSE method). (**a**) Geometric model and (**b**) FE model.

Figure 18 reveals that under different Mach numbers, the transmission loss and absorption coefficient obtained by the IBNSE method agreed very well with those obtained by the FNSE method, which once again proves the correctness of the IBNSE method. The FNSE method consumed 10 h and 20 min per calculation (corresponding to one Mach number), while the IBNSE method consumed only 3 h and 5 min per calculation, which demonstrates that the developed IBNSE method is computationally more efficient. It was observed that with the increase of incidence frequency, the transmission loss and acoustic absorption coefficient first increased and then decreased. The reason for this phenomenon can be explained by Figure 2h. As the frequency increases, the imaginary part of impedance of the perforated plate increases, which makes the imaginary part of the overall impedance change from negative to positive, with the point at which the imaginary part of the impedance is zero corresponds to the absorption peak. In addition, with the increase of Mach number, the peak values of transmission loss and absorption coefficient decreased, which led to a more uniform absorption performance within the considered frequency range. The reason is that an increase of Mach number leads to an increase in acoustic resistance and a decrease in acoustic reactance. The increased acoustic resistance usually results in a decreased peak value and increased bandwidth, while the decreased acoustic reactance shifts the absorption peak to a higher frequency.

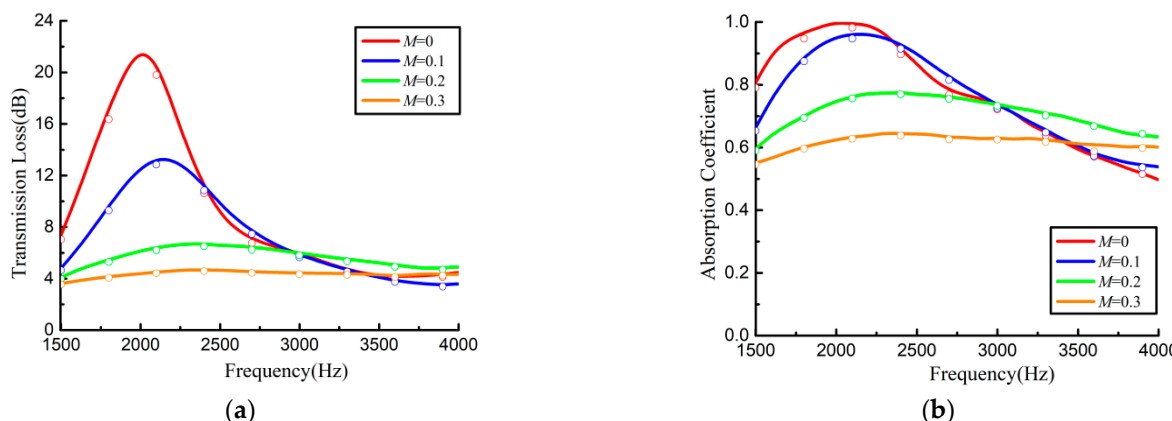

**Figure 18.** Comparisons of transmission loss and absorption coefficient of the RDPAA system under different grazing flow Mach numbers. The solid lines represent the results computed by the IBNSE method with Goodrich model, while the circles represent the results computed by the FNSE method. (**a**) Transmission loss and (**b**) acoustic absorption coefficient.

### 4.2. Annular Duct Acoustic Absorber

In addition to the rectangular duct, the planar BAU can be modified and applied to the annular duct structure. In Figure 19, the absorber with a circumferentially bent BAU is placed on the inner wall of the annular duct to form an annular duct–cylindrical acoustic absorber (ADCAA) system. In Figure 19, the outer and the inner radii of the duct

are $R_1 = 140.2$ mm and $R_2 = 110.2$ mm, respectively. The inner radius of the acoustic absorber is $r = 99.4$ mm and the outer radius is equal to $R_2$. The total length of the duct is $L_{\text{duct}} = 740$ mm and the total length of the cylindrical acoustic absorber is $L = 330$ mm.

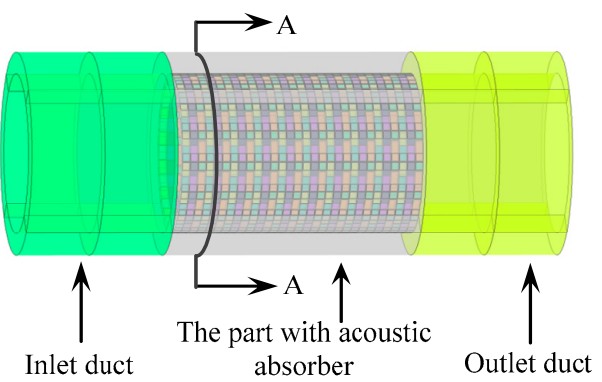

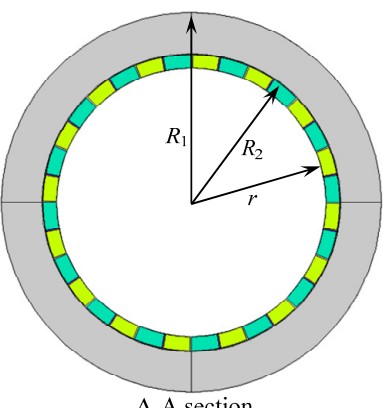

**Figure 19.** The ADCAA system.

In order to adapt the planar BAU to the cylindrical surface, the perforated plate and the cavities in the planar BAU were bent along the circumference of the duct, while the volume of each cavity was kept unchanged, as shown in Figure 20. The bent BAU structure constitutes the basic acoustic absorption unit of the cylindrical non-uniform acoustic absorber. In Figure 20, the inner radius $r = 99.4$ mm, the central angle of the bended BAU was $\theta = 21.95°$ and the axial length was $l = 34$ mm. The central angles corresponding to the six cavities were 15.37°, 11.52°, 10.98°, 10.43°, 9.88° and 6.04°. The other parameter values were consistent with the original planar BAU structure.

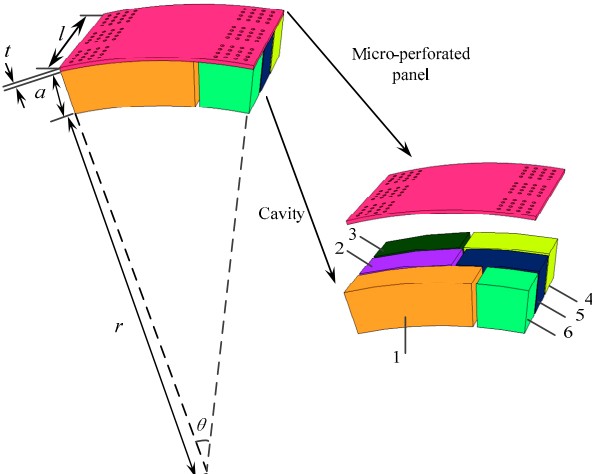

**Figure 20.** The bent BAU as the basic building block of the cylindrical non-uniform acoustic absorber.

The final cylindrical acoustic absorber was constructed by arranging 16 and 10 bent BAUs along the circumferential and axial directions, respectively, as shown in Figure 21. Then, the cylindrical absorber was placed in the inner wall of the annular duct to form an ADCAA system. The FE models that are suitable for the FNSE method and the IBNSE method are shown in Figures 22 and 23, respectively, in which only 1/4 of the cylindrical structure with symmetric boundary conditions was modeled.

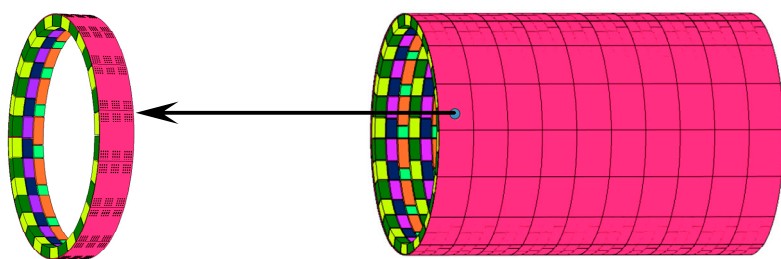

**Figure 21.** The cylindrical non-uniform acoustic absorber formed by the bent BAUs.

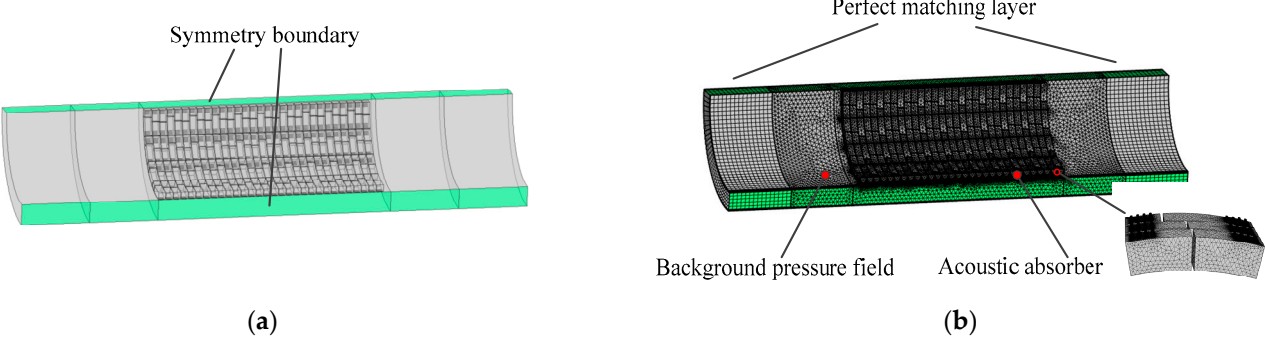

**Figure 22.** The geometric and detailed FE models of the ADCAA system with grazing flow (suitable for the FNSE method). (**a**) Geometric model and (**b**) FE model.

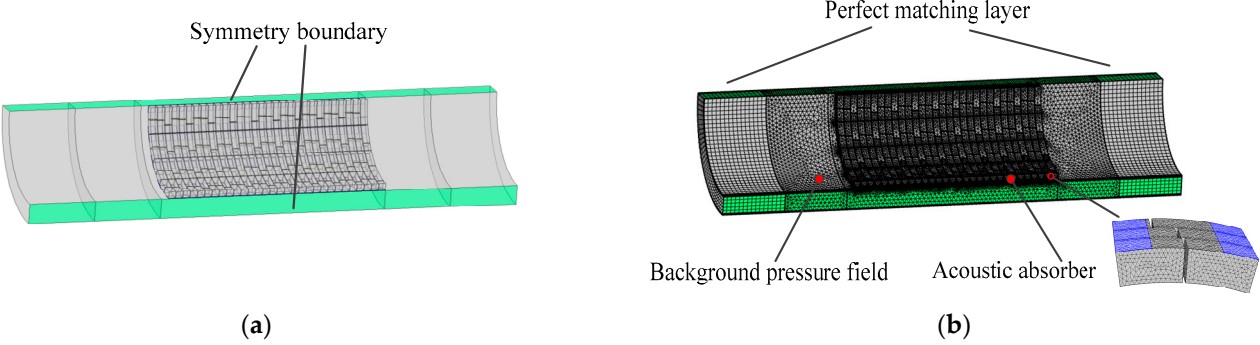

**Figure 23.** The geometric and detailed FE models of the ADCAA system with grazing flow (suitable for the IBNSE method). (**a**) Geometric model and (**b**) FE model.

It can be seen form Figure 24 that under different Mach numbers, the transmission loss and absorption coefficient obtained by the IBNSE method also agreed very well with those obtained by the FNSE computations. The designed cylindrical acoustic absorber had good absorption performance over a very wide frequency range (absorption coefficient was higher than 50%). The absorption characteristics of the ADCAA system were similar to those of the RDPAA structure. However, the influence of Mach number on the absorption coefficient of the ADCAA system was smaller than that of the RDPAA system, which resulted in relatively flat absorption curves for different Mach numbers. This reveals that compared with RDPAA system, the absorption performance of the ADCAA system is not sensitive to the variations of Mach number. In this example, the FNSE method consumed 25 h and 40 min per calculation, while the IBNSE method consumed 8 h and 30 min per calculation, demonstrating that the calculation efficiency was greatly improved.

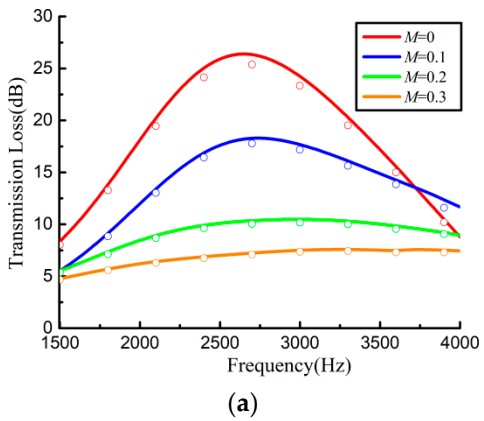
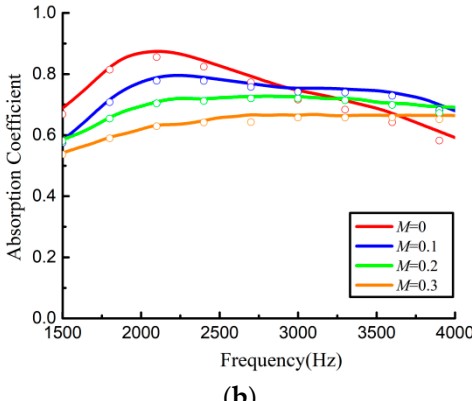

**Figure 24.** Transmission loss and absorption coefficient of the ADCAA system under different grazing flow Mach numbers. The solid lines represent the results computed by the IBNSE method with Goodrich model, while the circles represent the results computed by the FNSE method. (**a**) transmission loss. (**b**) acoustic absorption coefficient.

In terms of the manufacturing of the absorber proposed in this paper, titanium alloy materials can be used for manufacturing if the application scenario involves high temperatures or high stress. If the above cases are not involved, ABS (acrylonitrile butadine styrene) materials can be used to save costs. The whole structure can be fabricated by 3D printing.

## 5. Conclusions

In this paper, a broadband BAU absorber was first constructed by using an MPP and backed cavities with dissimilar lengths to produce peak absorption at multiple frequencies. Since each cavity length is extended along the MPP surface, different cavity volumes can be adopted to yield different resonance frequencies without increasing the thickness of the structure. The IBNSE method was developed to predict the attenuation characteristics of the duct acoustic system, in which comparisons of four semi-empirical impedance models were performed. It was found that the IBNSE method with Goodrich model was sufficient to accurately predict the acoustic attenuation of the absorber under the grazing flow condition. Using the BAU structure as the basic building block, planar and the cylindrical broadband non-uniform acoustic absorbers were constructed. The acoustic attenuation characteristics of the RDPAA and ADCAA systems under different grazing flow Mach numbers were calculated using the FNSE method as well as the developed IBNSE method. It was demonstrated that under different Mach numbers, these two acoustic systems exhibited good broadband absorption performances. The IBNSE method with Goodrich model was accurate and computationally efficient, and can be used to predict the absorption characteristics of acoustically treated ducts in the presence of grazing flow.

**Author Contributions:** All authors were involved in the study design, data acquisition, data analysis and data interpretation. All authors have read and agreed to the published version of the manuscript.

**Funding:** This research received no external funding.

**Institutional Review Board Statement:** Not applicable.

**Informed Consent Statement:** Not applicable.

**Data Availability Statement:** The data presented in this study are available on request from the corresponding author.

**Conflicts of Interest:** The authors declare no conflict of interest.

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
