# Peer review of "Prediction of the Absorption Characteristics of Non-Uniform Acoustic Absorbers with Grazing Flow"

_applsci, doi:10.3390/app13042256_

Round 1

Reviewer 1 Report

The submitted paper presents a numerical approach to design broadband acoustic absorbers for flow ducts. The basic absorption units consist of cavity-backed microperforated plates, multiple units with different dimensions are used to extend the frequency range. The basic idea of the presented approach is increasing the computational efficiency by employing semi-empirical impedance model of the microperforated plates instead of their detailed resolving and numerical modelling. This approach is not new in acoustics, and it is widely used in practice. The authors demonstrate that replacing the detailed model of the microperforated plate by its semiempirical impedance model reduces the computational cost of the numerical calculation, retaining very reasonable accuracy compared to the detailed model.

It is necessary to stress though that the authors compare two numerical models with each other, which is quite different thing that comparison with a real-world experiment. I think that if it is not possible for them to conduct their own experimental validation, they should benchmark their numerical results against previously published experimental results, to be able to claim in the Conclusions that their model can predict the acoustic attenuation of the absorber accurately under the grazing flow conditions. As it has been mentioned above, the authors compare results of two models.

Other issues:

1.       There is a lot of grammar issues, the manuscript should be checked by an English speaker.

2.       Line 139: Want does it mean that the sources are zero by default? Are the authors talking about mathematical model of physical reality, or preset values of these variables in COMSOL Multiphysics, which does not have to reflect a physical reality in a given context?

3.       Line 141: Equation (2a) is not the energy equation, it is linearized state equation of an ideal gas.

4.       Line 148: The authors should give the meaning of the coefficient beta_s?

5.       Line 151: The authors could mention the fact that the proportionality ratio is one over the sound speed squared.

6.       Paragraph beginning at line 164: The authors only mention the solution of turbulence equations without giving any details. In fact, in order to predict the absorber performance properly, the flow properties must be determined adequately. The authors should describe, for example,  the adopted model of turbulence and the initial conditions for the flow field. Are the authors sure that the flow pattern is fully developed at the position of the absorber?

7.       Figure 2: The impedances are either normalized with respect to some quantity (characteristic impedance), or they should have a unit.

8.       Section 42. Should probably not be called “Rectangular…”

Author Response

Thank you for your comments concerning our manuscript entitled ”Prediction of the absorption characteristics of the non-uniform acoustic absorber with grazing flow”. The comments are all valuable and very helpful for revising and improving our paper, as well as the important guiding significance to our researches. We have studied the comments carefully and have made correction. The main corrections in the paper and the responds to the comments are as following:

Comment: The submitted paper presents a numerical approach to design broadband acoustic absorbers for flow ducts. The basic absorption units consist of cavity-backed microperforated plates, multiple units with different dimensions are used to extend the frequency range. The basic idea of the presented approach is increasing the computational efficiency by employing semi-empirical impedance model of the microperforated plates instead of their detailed resolving and numerical modelling. This approach is not new in acoustics, and it is widely used in practice. The authors demonstrate that replacing the detailed model of the microperforated plate by its semiempirical impedance model reduces the computational cost of the numerical calculation, retaining very reasonable accuracy compared to the detailed model.

It is necessary to stress though that the authors compare two numerical models with each other, which is quite different thing that comparison with a real-world experiment. I think that if it is not possible for them to conduct their own experimental validation, they should benchmark their numerical results against previously published experimental results, to be able to claim in the Conclusions that their model can predict the acoustic attenuation of the absorber accurately under the grazing flow conditions. As it has been mentioned above, the authors compare results of two models.

Response: Special thanks to you for your good comments. The results obtained by the IBNSE method in this paper were compared with the experimental results given in Reference [15] (Figure 6). These comparisons are supplemented in Section 3.4 in the revised version. The comparison results show that the transmission loss and acoustic absorption coefficient obtained by the proposed method are in good agreement with the published experimental results, which indicates the accuracy of the proposed method.

Other issues:

  1. Comment:There is a lot of grammar issues, the manuscript should be checked by an English speaker.

Response: We are sorry for the grammatical errors and the manuscript has been further reviewed and corrected. We hope it meets the requirements.

  1. Comment:Line 139: Want does it mean that the sources are zero by default? Are the authors talking about mathematical model of physical reality, or preset values of these variables in COMSOL Multiphysics, which does not have to reflect a physical reality in a given context?

Response: The sources are preset values in COMSOL Multiphysics and are zero by default. The mass source term, force source term, and heat source term indicate that the system is not subjected to external mass, force, or heat.

  1. Comment:Line 141: Equation (2a) is not the energy equation, it is linearized state equation of an ideal gas.

Response: We are very sorry for our mistake and it is corrected to linearized state equation of an ideal gas in the revised version (see line 148).

  1. Comment:Line 148: The authors should give the meaning of the coefficient beta_s?

Response: Special thanks to you for your good comments. beta_s is the compression coefficients in adiabatic case, which is added in the revised version (see line 157).

  1. Comment:Line 151: The authors could mention the fact that the proportionality ratio is one over the sound speed squared.

Response: Special thanks to you for your good comments.  It is added in the revised version (see line 157).

  1. Comment: Paragraph beginning at line 164: The authors only mention the solution of turbulence equations without giving any details. In fact, in order to predict the absorber performance properly, the flow properties must be determined adequately. The authors should describe, for example,  the adopted model of turbulence and the initial conditions for the flow field. Are the authors sure that the flow pattern is fully developed at the position of the absorber?

Response: Special thanks to you for your good comments. The details of the turbulence equations are added in the revised version (see line 169). The background mean flow is modeled with a SST RANS model from the CFD Module in COMSOL Multiphysics. Based on the comparison between the simulation and experimental results, high accuracy can be achieved by considering the fluid near the absorber as a fully developed flow.

  1. Comment: Figure 2: The impedances are either normalized with respect to some quantity (characteristic impedance), or they should have a unit.

Response: Special thanks to you for your good comments. The impedances are modified to the relative surface impedance (unit 1), and to explain what it means, Eq. (23) is added in the revised version (see line 270).

  1. Comment: Section 4. Should probably not be called “Rectangular…”

Response: Special thanks to you for your good comments. In the title of Section 4.2, the word “Rectangular” is changed to “Annular” in the revised version (see line 488).

Reviewer 2 Report

In the first paragraph of introduction section authors introduced the metamaterial absorbers. Is the proposed absorber a metamaterial absorber if so a proper literature survey should be provided for metamaterial absorbers. And if it is not a metamaterial absorber then the relevance of this sentence should be discussed.

Proper referencing should be provided for each equation.

The relevance of Fig. 2 should be provided.

The comparison with available literature should be provided in form of comparison table.

fabrication techniques of the proposed absorber should be discussed.

The reason behind selecting this specific structure should be discussed and its importance and ease of fabrication should also be discussed.

If the proposed structure is Metamateiral absorber then author should citre the following literature:

Scientific Reports12(1), 18044. DOI: 10.1038/s41598-022-22951-1

Sustainability14(7), 4218. DOI: 10.3390/su14074218

Author Response

Thank you for your comments concerning our manuscript entitled ”Prediction of the absorption characteristics of the non-uniform acoustic absorber with grazing flow”. The comments are all valuable and very helpful for revising and improving our paper, as well as the important guiding significance to our researches. We have studied the comments carefully and have made correction. The main corrections in the paper and the responds to the comments are as following:

  1. Comment: In the first paragraph of introduction section authors introduced the metamaterial absorbers. Is the proposed absorber a metamaterial absorber if so a proper literature survey should be provided for metamaterial absorbers. And if it is not a metamaterial absorber then the relevance of this sentence should be discussed.

Response: Special thanks to you for your good comments. The proposed structure is not strictly a metamaterial absorber. In the introduction, the content about metamaterials is moved to the second paragraph to be introduced as one type of absorber in the revised version (see line 39).

  1. Comment: Proper referencing should be provided for each equation.

Response: Special thanks to you for your good comments. Reference [39] is added for Eq. (2-4) in the revised version (see line 148), Reference [15] is added for Eq. (7-8) in the revised version (see line 187).

  1. Comment: The relevance of Fig. 2should be provided.

Response: The relevance of Fig. 2 is added in the revised version (see line 287). Since the sound absorption performance of the absorber depends on the sum of the MPP impedance and the cavity impedance and there is no exact theoretical formula for the impedance of complex cavities in grazing flow, the acoustic absorption characteristics of absorbers with different parameters is not convenient to directly compare from Fig. 2. Therefore, the effects of each parameter in Fig. 2 are presented later in the manuscript.

  1. Comment: The comparison with available literature should be provided in form of comparison table.

Response: Special thanks to you for your good comments. The comparison with existing experimental results is supplemented in Section 3.4, and the comparison table “Table 2 and 3” is added in the revised version (see line 419). As demonstrated in Table 2 and 3, compared with the experimental results, the relative error of IBNSE method is similar to that of FNSE method. Therefore, the IBNSE method proposed in this paper has satisfactory computational accuracy.

  1. Comment: Fabrication techniques of the proposed absorber should be discussed.

Response: Fabrication techniques of the proposed absorber is added in the revised version (see line 537). The whole structure can be manufactured by 3D printing. The material can be resin or titanium alloy.

  1. Comment: The reason behind selecting this specific structure should be discussed and its importance and ease of fabrication should also be discussed.

Response: This structure was chosen because it uses different lengths of cavity arranged horizontally, which simultaneously has the characteristics of ultra-thin and wide-band sound absorption. Because the absorber does not contain complex internal structure, it can be machined directly by 3D printing and is easy to process.

  1. Comment: If the proposed structure is Metamateiral absorber then author should cite the following literature:

Scientific Reports, 12(1), 18044. DOI: 10.1038/s41598-022-22951-1

Sustainability, 14(7), 4218. DOI: 10.3390/su14074218

Response: Special thanks to you for your good comments. The literature is very valuable and cited as References [7,8] in the revised version (see line 39).

Reviewer 3 Report

it is not clear if the authors have provided any original metodological/formulation contribution. it seems they have  adopted available formulations to properly solve an interesting problem.

Author Response

Thank you for your comments concerning our manuscript entitled ”Prediction of the absorption characteristics of the non-uniform acoustic absorber with grazing flow”. The comments are all valuable and very helpful for revising and improving our paper, as well as the important guiding significance to our researches. We have studied the comments carefully and have made correction. The main corrections in the paper and the responds to the comments are as following:

Comment: It is not clear if the authors have provided any original methodological/formulation contribution. It seems they have adopted available formulations to properly solve an interesting problem.

Response: Special thanks to you for your good comments. The manuscript has been further reviewed and corrected. In this paper, the semi-empirical impedance model formulas are used to calculate the acoustic absorption performance of complex non-uniform acoustic absorber under grazing flow, which is rarely studied. In this paper, the comparison between the semi-empirical formulas and the FNSE method verifies that the Goodrich model has the highest accuracy, which can be used to improve the computational efficiency.

Round 2

Reviewer 1 Report

The authors responded to my question in a sufficient way, they have incorporated corrections to the issues raised within the previous review, I think the paper can be published as it is.

Reviewer 2 Report

The comments are satisfied. the paper can be accepted in its current form.